# Effects of scent lure on camera trap detections vary across mammalian predator and prey species

Dacyn Holinda[1]⊙, Joanna M. Burgar[1]⊙, A. Cole Burton[1,2]⊙ *

**1** Department of Forest Resources Management, Wildlife Coexistence Lab, University of British Columbia, Vancouver, British Columbia, Canada, **2** Biodiversity Research Centre, University of British Columbia, Vancouver, British Columbia, Canada

⊙ These authors contributed equally to this work.
* cole.burton@ubc.ca

**Data Availability Statement:** Data from this study were provided by a third party, the Alberta Biodiversity Monitoring Institute (ABMI), under a Data Sharing Agreement that does not permit the authors to redistribute the data. Camera trap

## Abstract

Camera traps are a unique survey tool used to monitor a wide variety of mammal species. Camera trap (CT) data can be used to estimate animal distribution, density, and behaviour. Attractants, such as scent lures, are often used in an effort to increase CT detections; however, the degree which the effects of attractants vary across species is not well understood. We investigated the effects of scent lure on mammal detections by comparing detection rates between 404 lured and 440 unlured CT stations sampled in Alberta, Canada over 120 day survey periods between February and August in 2015 and 2016. We used zero-inflated negative binomial generalized linear mixed models to test the effect of lure on detection rates for a) all mammals, b) six functional groups (all predator species, all prey, large carnivores, small carnivores, small mammals, ungulates), and c) four varied species of management interest (fisher, *Pekania pennanti*; gray wolf, *Canis lupus*; moose, *Alces alces*; and Richardson's ground squirrel; *Urocitellus richardsonii*). Mammals were detected at 800 of the 844 CTs, with nearly equal numbers of total detections at CTs with (7110) and without (7530) lure, and variable effects of lure on groups and individual species. Scent lure significantly increased detections of predators as a group, including large and small carnivore sub-groups and fisher specifically, but not of gray wolf. There was no effect of scent lure on detections of prey species, including the small mammal and ungulate sub-groups and moose and Richardson's ground squirrel specifically. We recommend that researchers explicitly consider the variable effects of scent lure on CT detections across species when designing, interpreting, or comparing multi-species surveys. Additional research is needed to further quantify variation in species responses to scent lures and other attractants, and to elucidate the effect of attractants on community-level inferences from camera trap surveys.

detection data may be obtained directly from the ABMI website (https://abmi.ca/home/data-analytics/da-top/da-product-overview/remote-camera-mammal-data/remote-camera-mammal-data-download.html). The ABMI does not release the exact locations of their monitoring locations (as part of their policy designed to protect the integrity of their long-term monitoring sites), but the data on habitat and disturbance at camera stations can be obtained by request from the ABMI (https://abmi.ca), specifically the Director of the Information Centre (Tara Narwani, 780-492-5531, tnarwani@ualberta.ca, https://www.abmi.ca/home/contact-us/abmi-directory.html).

**Funding:** This research was supported by the Canada Research Chairs Program.

**Competing interests:** The authors have declared that no competing interests exist.

## Introduction

Developing, testing, and refining methods for monitoring animal populations is increasingly important for wildlife researchers and managers around the world. As a result of human-driven habitat degradation, overharvest, climate change, and pollution, there is a consistent trend of declining biodiversity on a global scale [1,2]. In response to these biodiversity declines, there is a need for conservation practices that promote population recovery and stability for threatened and vulnerable species. A key challenge to evidence-based conservation is reliable assessment of wildlife distribution, abundance, and behaviour [3]. Gaps in knowledge of the status of many species and populations hinders our ability to assess extinction risk and improve conservation effectiveness [4]. The challenges of collecting robust data on wildlife populations and communities are therefore important to surmount [5].

Camera traps (CTs) are an increasingly common survey tool used in the monitoring of terrestrial vertebrate species, particularly those that are rare or elusive [6,7]. CTs record animals remotely and noninvasively, thus avoiding limitations and potential biases common to more invasive or targeted survey methods that require capture or direct observation of study species [6]. Data from CT surveys can be used to estimate population density, occupancy and behaviour for a wide range of animals [7–11], and these estimates can be used to directly inform management practices [12,13]. CT surveys also capture images of incidental (non-target) species, giving the opportunity for a single survey to provide data on a broader wildlife community with minimal additional sampling costs [7,9].

Despite the many advantages and applications of CTs, the method is not without its challenges. In order for inferences from CT data to be reliable, large numbers of animal detections are typically required [7,14]. However, CTs have relatively small detection zones, and many species targeted by CT surveys are wide-ranging, rare, or elusive, which often leads to a low number of detections [15]. Accordingly, CT surveys often rely on attractants, such as bait or scent lure, to increase detection probabilities. Baits are typically some form of food reward, such as a carcass or piece of meat, while lures are typically non-reward pastes or oils with a pungent odour [16]. While many opinions and experiences exist, the specific effects of attractants on CT detections is poorly studied [17,18].

There is thus a need to better quantify the effects of attractants on CT detections across species, and, particularly, to assess the potential for biased inferences. There is evidence that attractants alter movement patterns of some mammal species [19], which could affect inferences on species distribution, abundance, or habitat use [20]. For single-species surveys, the increased number of CT detections due to an attractant may be relatively more important than any change in movement behaviour [21]; however, the potential for bias may be magnified in multispecies surveys, as attractants are likely to have different effects on detectability for different species. Similarly, comparisons of results across studies that vary in their use of attractants may not be reliable. Since the effects of attractants are rarely considered in statistical analyses of CT data, there is considerable potential for biases in estimates of population and community attributes and thus in the management recommendations they inform.

Our study aimed to test whether the presence of a scent lure influenced overall CT detection rates of mammals, and whether this influence was consistent across different mammal species and groups. We capitalized on an extensive CT sampling program in Alberta, Canada, that included both lured and unlured camera stations. We hypothesized that behavioural differences between mammalian predator and prey species would lead to differential responses to a pungent scent lure typical of CT surveys. More specifically, we hypothesized that scent lure would increase CT detections for mammalian predators, but decrease detections for prey species likely to be warier of smells mimicking animal carcasses.

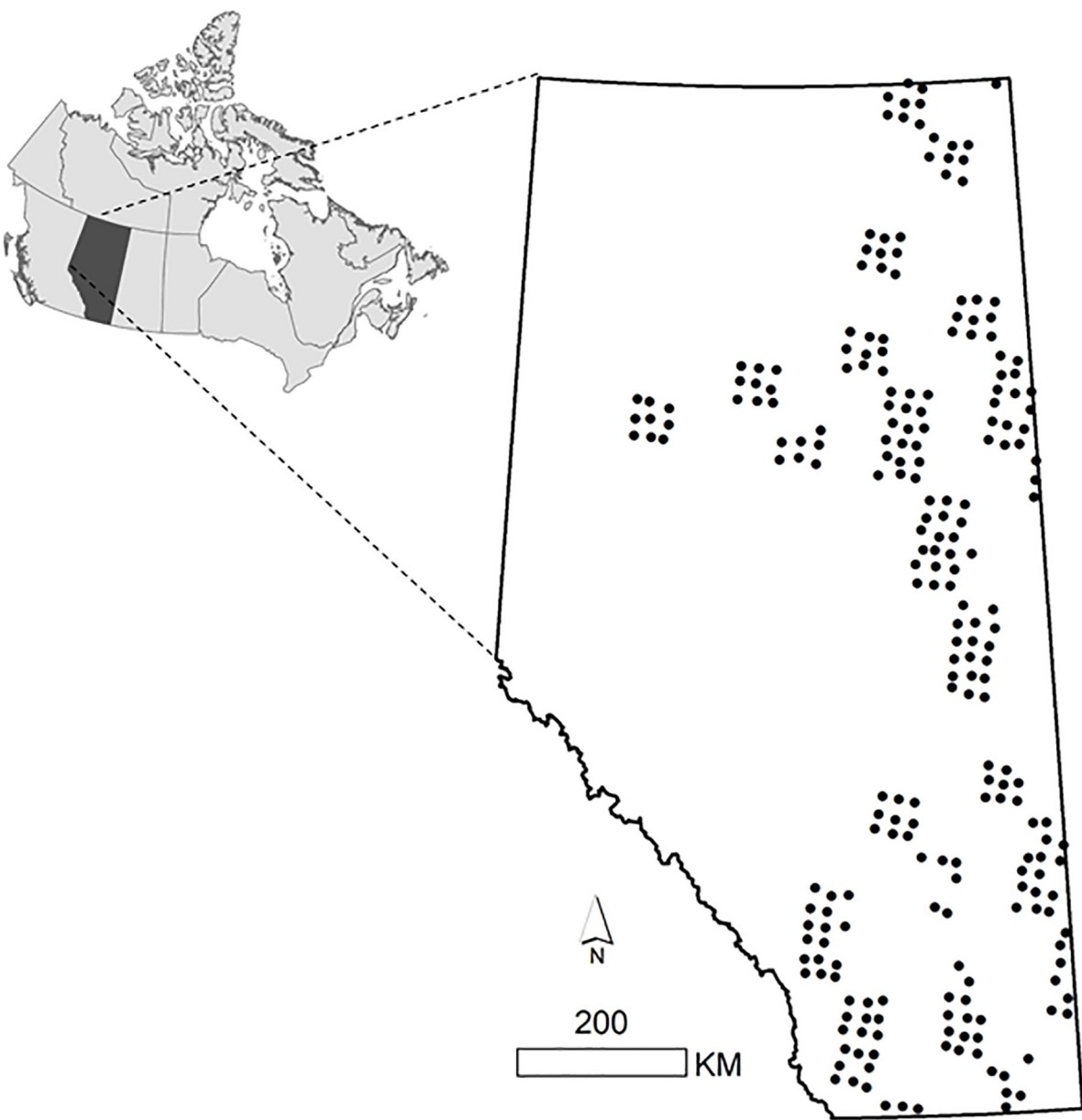

**Fig 1. Camera trap sites (n = 247) from the Alberta Biodiversity Monitoring Institute used to assess the effects of scent lure on mammal detections.** Each dot represents a monitoring site comprised of four camera traps (2 lured, 2 unlured; 844 in total) that were sampled for 120 consecutive days between February and August, 2015 or 2016, in Alberta, Canada (inset).

## Materials and methods

Data were collected from CT survey sites deployed across Alberta (Fig 1) as part of a monitoring program implemented by the Alberta Biodiversity Monitoring Institute (ABMI; www.abmi.ca). CTs were deployed between February and August in 2015 and 2016 at sites spaced 20 km apart in a systematic grid based on Canada's National Forest Inventory [22] (CT sites extended from 49.05 to 60.00 decimal degrees latitude, and -110.14 to -117.67 longitude; Fig 1). Four Reconyx PC900 CTs (Reconyx, Holmen, Wisconsin) were deployed at each site, spaced 600 m apart at the corners of a square surrounding the central grid point (Fig 2; details

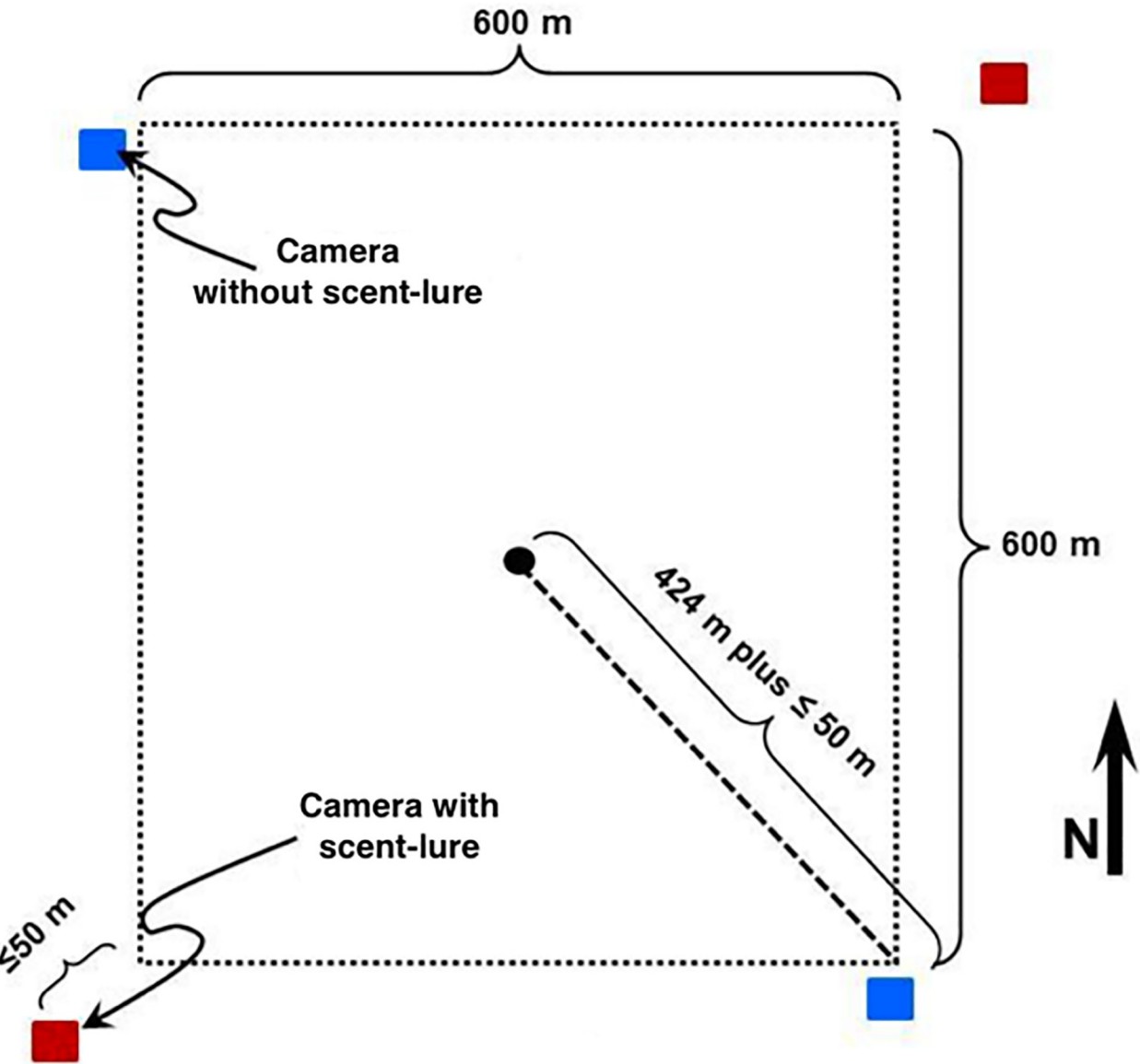

**Fig 2. Placement of camera traps at an ABMI terrestrial monitoring site (diagram modified from [22]).** The central black dot represents the ABMI site centre, within their systematic provincial grid (Fig 1). Red squares represent camera traps with scent lures, located at the northeast and southwest corners of the site. Blue squares represented camera traps without a scent lure, located at the northwest and southeast corners of the site.

in [22]). CTs located in the northeast and southwest corners were deployed with one table-spoon of O'Gorman's Long Distance Call (O'Gorman Long Line Lures, Broadus, Montana), a scent lure with a high skunk and musk blend, typical of many CT surveys. The lure was placed inside a polyvinyl chloride (PVC) tube fastened to the ground 5 m in front of the camera by a stake. No lure was placed at the cameras in the northwest and southeast corners of the site. To ensure that the potential influence of lure was comparable across CT stations with equal sampling effort, we only included CTs that were active continuously and we truncated data to 120 days post deployment. Our final dataset included 844 CT stations at 247 ABMI monitoring sites. In 2015, scent lure was present at 206 cameras and absent at 236 cameras, compared to

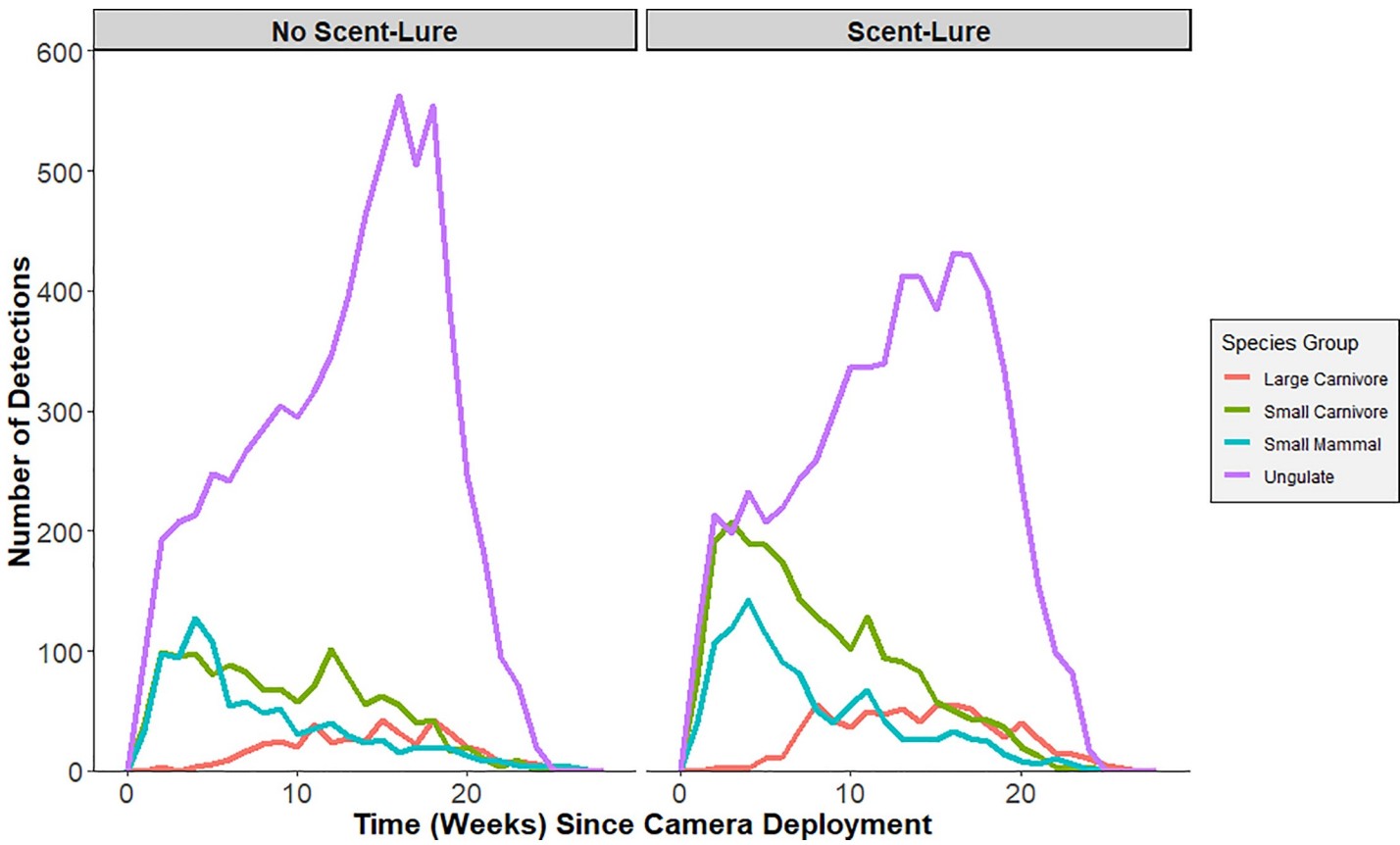

**Fig 3. Number of mammal detections by time since camera deployment.** Detections are distinguished between species groups and between camera traps deployed with scent lure (n = 10480 detections) and without scent lure (n = 9774). Statistical analyses of the effect of lure only used detections occurring within the first 17 weeks (120 days) since deployment.

198 and 204, respectively, in 2016. Sites were sampled in either 2015 or 2016, but not both years.

ABMI staff classified species detections from images recorded by the CTs. To reduce the influence of animals spending time in front of the camera (e.g. investigating lure), we considered minimum time intervals of 5, 10, 30 and 60 min between successive images of the same species at the same site as potential criteria for defining independent detection events [11]. As we found little sensitivity in the number of events across these intervals, we used the 30 min threshold to define events for subsequent analyses. We excluded domesticated mammal species from the detection dataset (S1 Table). The remaining wild mammal species were classified into functional groupings as either predators—with sub-groups of large (>20 kg) and small (<20 kg) carnivores—or prey, with sub-groups of ungulates and small mammals (S1 Table). We chose one species of management concern with a moderate number of detections from each sub-group to assess whether the group-level results were generalizable to the species level; specifically: gray wolf *Canis lupus*, fisher *Pekania pennanti*, Richardson's ground squirrel *Urocitellus richardsonii*, and moose *Alces alces*. We made the simplifying assumption that any effect of scent lure would last for the 120 day sampling period and used the total detections for each species or group as our response variable. Research into scent lure longevity for stoat (*Mustela spp.*) suggested that commercially available scent lures are longer lasting than baits [23], and a visual inspection of detections over the 120 day sampling period showed similar trends in the frequency of detections for each species group, regardless of lure (Fig 3).

Aside from our main experimental treatment of lure vs. no lure (binary variable), we included two additional categorical variables to control for variation in detections due to habitat type and human disturbance around each CT. The ABMI monitoring program assessed habitat and disturbance within a 50 m radius around each camera location [22]. We aggregated field data into three broad habitat categories (forest, grassland, and wetland; S2 Table) and two classes of human disturbance (disturbed vs not disturbed; S3 Table).

We performed statistical analysis using R software version 3.3.2 [24]. We modelled CT detection events as a count response variable reflecting the local abundance, activity and detectability of mammal species around each CT. We developed zero-inflated negative binomial (ZINB) generalised linear mixed models (GLMMs) using the glmmTMB package [25], since the count data were non-normal, overdispersed (better fit by NB than by Poisson distribution), and generally included a high proportion of zeros reflecting where focal species or groups were not detected at CTs (results were similar whether or not the zero-inflation factor was included). We included site as a random effect to account for the clustering of four cameras at each site. Models were run using data subset to each group, sub-group and species, in addition to one overall model comprising all data. We first ran a set of candidate models on the full dataset of mammal detections, with the number of detections at a CT regressed against the following combinations of predictor variables: a) lure, habitat type and human disturbance (full additive model), b) lure and habitat type, c) lure and human disturbance, or d) lure only. Reference categories for these categorical variables were: no lure, forest, and no disturbance. Models were ranked by Akaike Information Criterion (AIC; [26]) and the top candidate model was selected as the one with the lowest AIC score, which was also significantly different from the next ranked model based on a Chi square test with α set to 0.05 (S4 Table). We then used this same top model and ran ZINB GLMMs for each subset of data: group (predators and prey), sub-group (large carnivores, small carnivores, small mammals and ungulates) and species (gray wolf, fisher, Richardson's ground squirrel, and moose). Due to small sample sizes and issues with model convergence, we revised the habitat covariate to a binary variable (vegetated vs. wetland) for the fisher and ground squirrel models and removed the zero-inflation from the wolf model.

## Results

Across the 844 CTs, mammals were detected at 800 CTs. There were 14,640 mammal detections: 3538 predator detections (813 large carnivore, 2725 small carnivore) and 11,102 prey detections (1507 small mammal, 9595 ungulate; Fig 3; S1 Table). The most frequently detected species were white-tailed deer (*Odocoileus virginianus*, 5278 detections), mule deer (*Odocoileus hemionus*, 2653), and coyote (*Canis latrans*, 2007), while bobcat (*Lynx rufus*) and wild boar (*Sus scrofa*) were each only detected once (Fig 4; S1 Table). There were nearly equal numbers of total detections at CTs with scent lure (7110 detections) compared to CTs without scent lure (7530) during the 120 sampling periods.

The top candidate model for all detections included effects of lure and habitat type (S4 Table; parameter estimates for all models given in S5 Table). Scent lure had an overall positive effect on mammal detections ($\beta$ = 0.29 ± 0.06 SE, $z$ = 4.82, $p$ < 0.001; Fig 5), and detections were lower in wetlands compared to forest ($\beta$ = -0.52 ± 0.10 SE, $z$ = -5.35, $p$ < 0.001). For models run on subgroups of species, predators and prey differed in their response to scent lure, with a positive effect of lure on predator detections ($\beta$ = 0.75 ± 0.07 SE, $z$ = 10.45, $p$ < 0.001) but no effect on prey detections ($\beta$ = 0.02 ± 0.07 SE, $z$ = 0.27, $p$ = 0.789). Scent lure also had a strong, positive effect on large and small carnivore detections ($\beta$ = 0.81 ± 0.13 SE and 0.78 ± 0.08, $z$ = 6.09 and 9.46, respectively; both $p$ < 0.001). At our finest level of taxonomic resolution, the two focal predator species differed in their responses, with scent lure having no

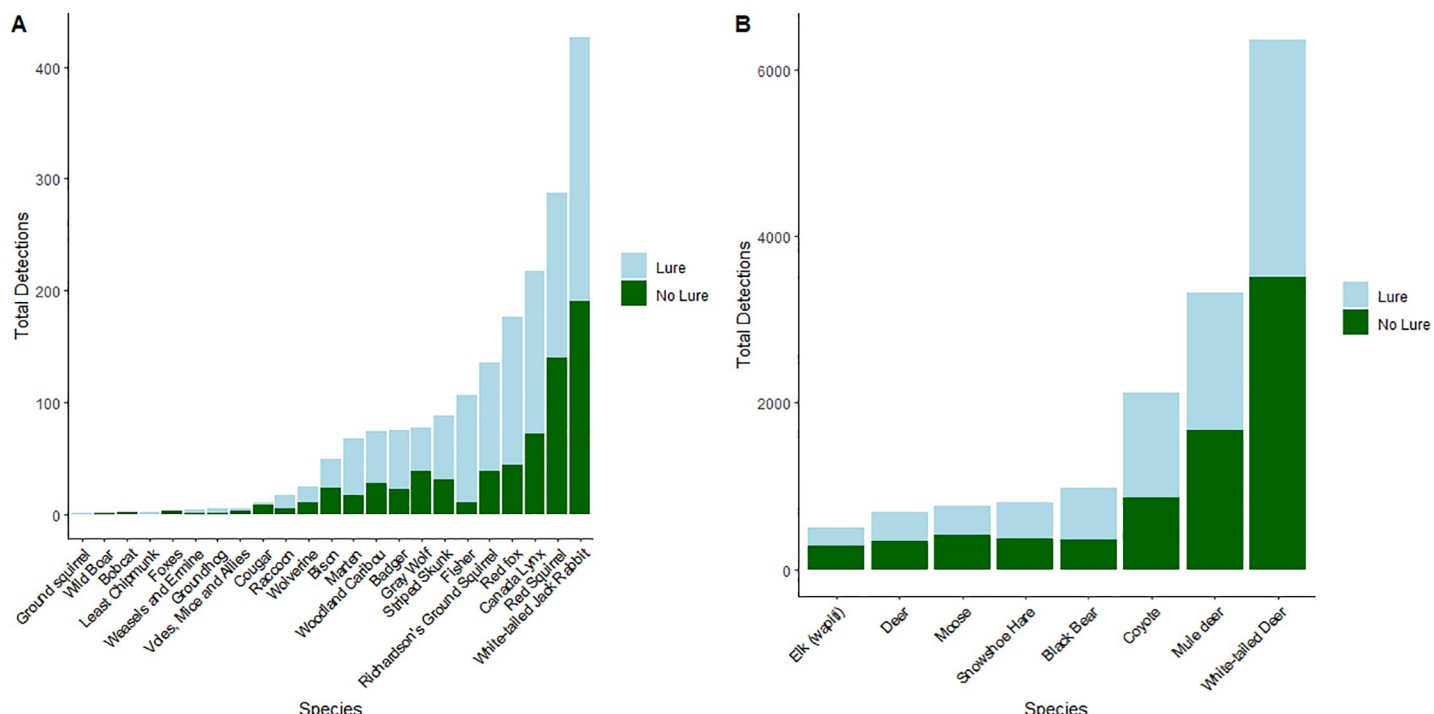

**Fig 4. Total camera trap detection events for each mammal species.** Camera traps were active for 120-day sampling periods between 17 Feb—25 Aug, 2015 and 16 Feb—23 Aug, 2016. Light blue shading represents detections at camera stations with scent lure (n = 404), while green shading represents detections at stations without scent lure (n = 440).

effect on detections of wolf ($\beta$ = -0.01 ± 0.34 SE, $z$ = -0.03; $p$ = 0.973) but a strong, positive effect on fisher ($\beta$ = 2.23 ± 0.36 SE, $z$ = 6.18; $p$ <0.001). The effect of scent lure was consistently negligible for prey groups, with no statistically significant effect on small mammal or ungulate detections ($\beta$ = 0.29 ± 0.17 SE and -0.03 ± 0.08, $z$ = 1.75 and -0.41, $p$ = 0.080 and 0.680, respectively), nor on detections of Richardson's ground squirrel or moose ($\beta$ = 0.00 ± 1.14 SE and 0.07 ± 0.15, $z$ = 0.00 and 0.50, $p$ = 0.998 and 0.621; respectively; Fig 5; S5 Table).

Temporal patterns of mammal detections across the period of CT sampling were similar between lured and unlured cameras, but displayed some interesting seasonal patterns (Fig 3). Carnivore detections were relatively consistent, with a slight increase during weeks 10–20, likely due to the spring emergence of black bear—there were only 5 detections of black bear in the first 30 days following CT deployment and all remaining detections in the latter 90 days of CT activity (Fig 3). Small carnivores and small mammals showed a peak in detections 2–4 weeks following deployment, regardless of scent lure deployment, and then a gradual decrease in detections. For ungulates, the largest number of detections occurred between weeks 10 and 20 of the CTs being active.

## Discussion

Scent lure was shown to have a positive impact on CT detections of predators as a group, including the predator subgroups and species assessed separately, with the exception of gray wolf. By contrast, prey species on the whole, and the subgroups of prey evaluated, did not show any response to lure (Figs 4 and 5). These results generally support our hypothesis that scent lure increases the number of CT detections of predator species, although the effect was not universal. The results also corroborate previous research suggesting that lures can increase

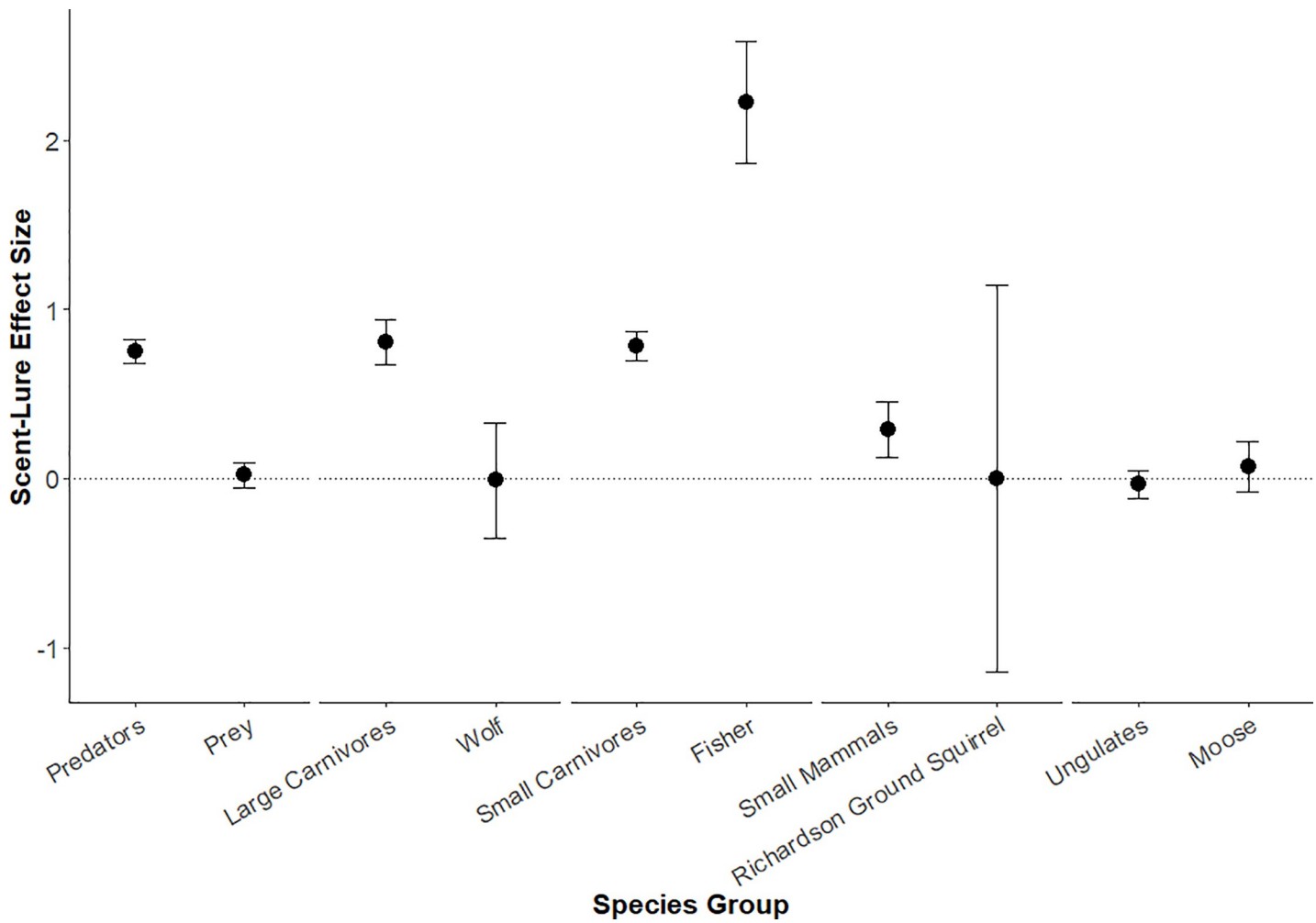

**Fig 5. Effect of scent lure on detections of different species and groups.** Mean effect (± SE) estimated from generalized linear mixed models of camera trap detections across lured (n = 404) and unlured (n = 440) stations in Alberta, Canada. Zero (no effect) is indicated by a horizontal dotted line.

CT detections of predator species [14,17]. The observed increase in detections of both large carnivores and small carnivores at lured CTs is likely explained by the fact that scent lures, such as the Long Distance Call lure used in this study, are targeted towards carnivores via scents associated with common prey species [16]. This scent likely draws investigative predator species into the CT detection area, increasing the total number of predator detections. Studies on black and brown bears corroborate this claim, showing that combined CT and hair snag sites with a scent lure detected a greater number of bears and collected more hair compared to sites without a scent lure [27]. More recent studies on CT attractants have also shown that the presence of a scent lure can influence predator behaviour such that they will alter their movement patterns in an attempt to find the source of the scent, which may lead to an increased number of predator detections at lured CTs [28]. Nevertheless, the absence of an effect on lure on wolf detections, contrasted by the particularly large effect on fisher detections, highlights the fact that variation in responses to lure among predator species (and perhaps individuals) should be carefully considered.

Contrary to our predictions, we did not find that prey avoided CTs with a scent lure. Small mammals and ungulates were neither attracted to nor avoided scent lures (Figs 4 and 5). Small

mammal detections were dominated by a few species, with over two-thirds of detections in this group being either snowshoe hare (*Lepus americanus*; 729 detections) or white-tailed jack rabbit (*Lepus townsendii*; 403 detections). Some previous research has suggested that snowshoe hares may be attracted to the scent of commercially available hunting lures [29]; thus, the weakly positive effect ($p$ = 0.08; Fig 5) of lure on small mammal detections that we observed may be more related to hares, rather than small mammals generally, especially considering the lack of an association between scent lure and Richardson's ground squirrel detections. Ungulate detections were dominated by white-tailed and mule deer, but similar patterns were observed for moose and elk (Fig 4B), implying the lack of a lure effect was general for this group.

Human disturbance and habitat features can significantly influence CT detections of a variety of mammal species [30,31]. Despite our relatively simple categorization of these factors, we also found a significant influence of habitat on mammal detections, with fewer detections in wetland compared to forest habitats. Our disturbance variable was not included in the top model for detections across all species, and thus was not further evaluated for sub-groups, but we note that both negative and positive effects of land use disturbances have been documented using CTs for several of these species (e.g. [32]). Continued research into CT methodology should consider interactions between habitat and CT attractants in relation to CT detections of various focal and non-focal species.

Ultimately our results suggest that scent lure has different effects on the detectability of different species by camera traps, including variation in both the direction and strength of effect. This result has important implications for multispecies CT surveys. While many CT studies focus specifically on multiple species [7], or make use of "bycatch" of non-target species (e.g. [33, 34]), the variable effects of the use of attractants like scent lure are rarely considered. This may introduce unknown bias into inferences across multiple species from CT surveys using attractants, or in comparisons between surveys that differed in their attractant use. Our results suggest that multispecies CT studies must account for the variable effect of scent lure on different species, and that greater statistical rigour is required when dealing with the effect of scent lure specifically, and attractants generally, as a means of influencing species CT detections in multispecies studies. At minimum, it will likely be important to incorporate the use of lure as a model covariate that can vary across sites, species, and time when comparing CT detections across studies and periods. Future research should investigate the ability of different modelling techniques, including occupancy models [35], to reliably account for variable effects of attractants on species detectability while estimating population and community parameters of interest.

Failure to adequately account for the variable influence of attractants may result in incorrect conclusions regarding species distribution, abundance or behaviour. Inferences about species interactions and other community dynamics may be particularly prone to bias when members of the community are responding in different ways to the sampling method. Furthermore, our results suggest the need for additional species-specific evaluation of the influence of scent lure on CT detections, within a variety of environmental and regional contexts, as well as research on the effects of different attractants. Researchers must strive to develop sampling and analytical frameworks that are robust to the potential biases of attractants, especially when comparing CT detections across species or between surveys using different methods.

## Supporting information

**S1 Table. Summary of camera trap detections.**
(PDF)

**S2 Table. Habitat types and their frequencies across camera trap stations.**
(PDF)

**S3 Table. Human disturbance types and their frequencies across camera trap stations.**
(PDF)

**S4 Table. AIC model selection for candidate generalized linear mixed models.**
(PDF)

**S5 Table. Parameter estimates for fixed effects from generalised linear mixed models.**
(PDF)

## Acknowledgments

Data for this study were provided by the Alberta Biodiversity Monitoring Institute (ABMI, www.abmi.ca). We thank all ABMI staff involved in the collection and dissemination of these data, with particular thanks to C. Copp, T. Narwani, and J. Schieck. Feedback on an early draft of this research was provided by E. Taylor and J. Myers.

## Author Contributions

**Conceptualization:** Dacyn Holinda, A. Cole Burton.

**Data curation:** Dacyn Holinda, Joanna M. Burgar.

**Formal analysis:** Dacyn Holinda, Joanna M. Burgar.

**Methodology:** Dacyn Holinda, Joanna M. Burgar, A. Cole Burton.

**Project administration:** A. Cole Burton.

**Supervision:** A. Cole Burton.

**Validation:** Joanna M. Burgar.

**Writing – original draft:** Dacyn Holinda, A. Cole Burton.

**Writing – review & editing:** Joanna M. Burgar, A. Cole Burton.

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
