## [Decision Letter · Decision Letter 0]

19 Mar 2020

PONE-D-20-02366

Effects of scent lure on camera trap detections vary across mammalian predator and prey species

PLOS ONE

Dear Dr Burton,

Thank you for submitting your manuscript to PLOS ONE. After careful consideration, we feel that it has great merit and will be acceptable following a few very minor edits and a few changes mostly related to the statistical analysis. As someone who uses this approach, I found you paper very well developed and likely to be of great use for those wishing to employ camera traps in their research. Therefore, we invite you to submit a revised version of the manuscript that addresses the points raised during the review process.

Reviewer #1 raises several important issues including data availability, statistical analyses, presentation of models and results,  and comparison to occupancy models. Please address these issues in your revision. 

We would appreciate receiving your revised manuscript by May 03 2020 11:59PM. To enhance the reproducibility of your results, we recommend that if applicable you deposit your laboratory protocols in protocols.io, where a protocol can be assigned its own identifier (DOI) such that it can be cited independently in the future. For instructions see: http://journals.plos.org/plosone/s/submission-guidelines#loc-laboratory-protocols

We look forward to receiving your revised manuscript.

Kind regards,

Tim A. Mousseau

Academic Editor

PLOS ONE

Journal Requirements:

2. In your Methods section, please provide additional location information of the study area, including geographic coordinates for the data set if available.

4. We note that Figure 1 in your submission contain map images which may be copyrighted. All PLOS content is published under the Creative Commons Attribution License (CC BY 4.0), which means that the manuscript, images, and Supporting Information files will be freely available online, and any third party is permitted to access, download, copy, distribute, and use these materials in any way, even commercially, with proper attribution. For these reasons, we cannot publish previously copyrighted maps or satellite images created using proprietary data, such as Google software (Google Maps, Street View, and Earth). For more information, see our copyright guidelines: http://journals.plos.org/plosone/s/licenses-and-copyright.

Reviewers' comments:

Reviewer's Responses to Questions

**Comments to the Author**

1. Is the manuscript technically sound, and do the data support the conclusions?

Reviewer #1: Yes

2. Has the statistical analysis been performed appropriately and rigorously? 

Reviewer #1: Yes

3. Have the authors made all data underlying the findings in their manuscript fully available?

Reviewer #1: No

4. Is the manuscript presented in an intelligible fashion and written in standard English?

Reviewer #1: Yes

5. Review Comments to the Author

Reviewer #1: This is an interesting paper that measures the effect of lures on the detection rate of multiple species and functional groups across a wide region of the province of Alberta in Canada. The paper is well written and the questions and methods are clearly described. I only have a few suggestions/comments:

- Can the authors provide a hyperlink pointing to the data set? I did not see in the main manuscript.

- Please justify the GLMM approach with a zero inflated binomial link.

- The camera traps within a site seem relatively close to each other for many of the species studied. Was spatial autocorrelation in detection rate modeled for each species?

- Please include a table with values of all parameters for each model, including standard errors. Some of the parameters are mentioned in the MS, but the information needs to be included in a table for easy inspection.

- What do the authors recommend people do when comparing between camera trap studies with different levels of lure or scent, specially when modeling population trends over time? Does it make sense to include lure as a covariate at the camera trap sample level per species? or functional group?

- Another possible approach is to model occupancy and detection probability for each species with lure as a covariate (and site as a random effect). What would the authors predict in this case? My suspicion is that lure affects the detection probability for certain species but not the underlying population metrics. Please discuss and put into a wider context.

Hope these comments are useful and congratulations for a great study!

6. PLOS authors have the option to publish the peer review history of their article (what does this mean?). If published, this will include your full peer review and any attached files.

Reviewer #1: No

---

## [Author Response · Author response to Decision Letter 0]

8 Apr 2020

Please see attached file "Response to Reviewers.docx"

---

## [Editor Report · Decision Letter 1]

27 Apr 2020

Effects of scent lure on camera trap detections vary across mammalian predator and prey species

PONE-D-20-02366R1

Dear Dr. Burton,

We are pleased to inform you that your manuscript has been judged scientifically suitable for publication and will be formally accepted for publication once it complies with all outstanding technical requirements.

With kind regards,

Tim A. Mousseau

Academic Editor

PLOS ONE
---

## [Editor Report · Acceptance letter]

1 May 2020

PONE-D-20-02366R1 

Effects of scent lure on camera trap detections vary across mammalian predator and prey species 

Dear Dr. Burton:

I am pleased to inform you that your manuscript has been deemed suitable for publication in PLOS ONE. Congratulations! Your manuscript is now with our production department. 

With kind regards,

on behalf of

Dr. Tim A. Mousseau 

Academic Editor

PLOS ONE